# Brain-specific inhibition of mTORC1 eliminates side effects resulting from mTORC1 blockade in the periphery and reduces alcohol intake in mice

Yann Ehinger [1,3], Ziyang Zhang [2,3], Khanhky Phamluong[1], Drishti Soneja[1], Kevan M. Shokat [2] & Dorit Ron [1✉]

Alcohol Use Disorder (AUD) affects a large portion of the population. Unfortunately, efficacious medications to treat the disease are limited. Studies in rodents suggest that mTORC1 plays a crucial role in mechanisms underlying phenotypes such as heavy alcohol intake, habit, and relapse. Thus, mTORC1 inhibitors, which are used in the clinic, are promising therapeutic agents to treat AUD. However, chronic inhibition of mTORC1 in the periphery produces undesirable side effects, which limit their potential use for the treatment of AUD. To overcome these limitations, we designed a binary drug strategy in which male mice were treated with the mTORC1 inhibitor RapaLink-1 together with a small molecule (RapaBlock) to protect mTORC1 activity in the periphery. We show that whereas RapaLink-1 administration blocked mTORC1 activation in the liver, RapaBlock abolished the inhibitory action of Rapalink-1. RapaBlock also prevented the adverse side effects produced by chronic inhibition of mTORC1. Importantly, co-administration of RapaLink-1 and RapaBlock inhibited alcohol-dependent mTORC1 activation in the nucleus accumbens and attenuated alcohol seeking and drinking.

[1] Department of Neurology, University of California San Francisco, San Francisco, CA, USA. [2] Department of Cellular and Molecular Pharmacology and Howard Hughes Medical Institute, University of California, San Francisco, CA, USA. [3]These authors contributed equally: Yann Ehinger, Ziyang Zhang. ✉email: dorit.ron@ucsf.edu

Alcohol use disorder (AUD) is characterized by compulsive alcohol intake despite negative consequences[1,2]. AUD is widespread, affecting 10–15% of the population, causing significant medical, social, and economic burdens[2–4]. In fact, AUD is one of the most prevalent mental health disorders[2], and the incidence of AUD diagnosis has increased by 35% in the United States between 2001 and 2013[1]. Unfortunately, pharmacotherapeutic options for treating AUD are limited, and only three drugs, naltrexone, acamprosate, and disulfiram, have been approved by the US Food and Drug Administration (FDA) as therapeutics for AUD[5]. Thus, there is a need to develop additional effective medications to alleviate phenotypes such as binge drinking, craving, and relapse.

The mechanistic target of Rapamycin complex 1 (mTORC1) represents a valuable drug target for the treatment of AUD. mTORC1 is a multiprotein complex that contains the serine/threonine-protein kinase mTOR and adaptor proteins, including Raptor, Deptor, and mLST8[6,7]. mTORC1 is activated by growth factors, amino acids, and oxygen[7,8], and has a role in lipid genesis, glucose homeostasis, protein translation[7,8], and autophagy[9]. Hyperactivation of mTORC1 has been linked to pathological states such as insulin resistance and cancer[7,8]. In the central nervous system (CNS), mTORC1 is activated by neurotransmitters and neuromodulators, such as glutamate and BDNF[6,10]. Upon activation, mTORC1 phosphorylates eIF4E-binding protein (4E-BP) and the ribosomal protein S6 kinase (S6K), which in turn phosphorylates its substrate, S6[7]. These phosphorylation events precede the initiation of local dendritic translation of synaptic proteins[11,12]. As such, mTORC1 has an important role in synaptic plasticity, and learning, and memory[6,13]. mTORC1 malfunction in the CNS has been linked to aging processes[14], neurodegenerative diseases such as Alzheimer's disease, Parkinson's disease, and Huntington's disease[8,13,15], neurodevelopmental disorders such as autism, as well as psychiatric disorders including addiction[13,14,16]. Growing evidence in rodents implicates mTORC1 in mechanisms underlying AUD[17]. Specifically, excessive alcohol drinking activates mTORC1 in the nucleus accumbens (NAc) and the orbitofrontal cortex of mice and rats[18,19]. The activation of mTORC1 by alcohol is specific and is not detected after the consumption of the natural rewarding substrate sucrose[19]. mTORC1 activation by alcohol is also brain-region specific as it is not observed in other striatal and cortical regions[19]. Using rapamycin, a selective allosteric mTORC1 inhibitor[20], and RapaLink-1, a third-generation mTORC1 inhibitor[21] in combination with rodent paradigms that model phenotypes associated with AUD, mTORC1 was found to contribute to mechanisms underlying alcohol-seeking, excessive alcohol consumption[18,22,23], alcohol reward, and habit[10,18,22]. Finally, mTORC1 also plays a crucial role in mechanisms that drive relapse to alcohol drinking[24,25], a crucial step in the addiction cycle[26].

Because of the important role of mTORC1 in various pathological states, the kinase represents an attractive drug target for the treatment of numerous diseases. Indeed, rapamycin and its analogs (rapalogs) have been approved by the FDA for the prevention of organ rejection after transplantation[27,28], as well as for the treatment of several types of cancer, tuberous sclerosis, and cardiovascular disease[29,30]. However, chronic inhibition of mTORC1 in the periphery produces detrimental side effects such as thrombocytopenia, impaired glucose sensitivity, hyperlipodemia, decreased wound healing, and the suppression of the immune system[29,31], thus limiting the utility of rapamycin and other rapamycin-derivatives (rapalogs) for the treatment of CNS disorders such as AUD because of safety concerns.

In an attempt to circumvent these undesirable effects resulting from sustained mTORC1 inhibition in the periphery, we developed an approach, which enables CNS-specific inhibition of mTORC1 while protecting the activity of the kinase in the periphery[32] (Fig. 1). Specifically, we utilized the unique mechanism of action of the mTORC1 inhibitors, rapamycin, and RapaLink-1, which requires their binding to the chaperone, FK506 binding protein 12 (FKBP12) prior to the inhibition of the kinase[33] (Fig. 1, Supplementary Fig. 1). We designed a brain impermeable small molecule (RapaBlock) that binds FKBP12, and acts to prevent access to the necessary factor for mTOR inhibition (Fig. 1, Supplementary Fig. 1). We hypothesized that when RapaBlock will be co-administered with RapaLink-1, mTORC1 activity will be protected in the periphery while inhibited in the brain[32] (Fig. 1). We further predicted that this approach would block the undesirable side effects observed after chronic inhibition of the kinase. Finally, we tested the utility of the approach in a preclinical mouse model of AUD.

## Results

**Rapablock blocks Rapalink-1 inhibition of mTORC1 in the liver.** First, to determine whether RapaBlock protects mTORC1

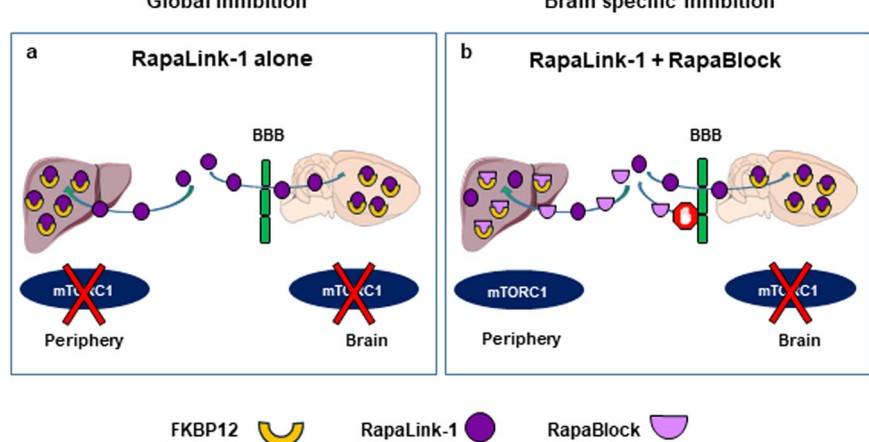

**Fig. 1 Schematic representation of strategy. a** Systemic administration of RapaLink-1 (purple) inhibits mTORC1 in the periphery and in the brain. **b** RapaBlock (pink), a small molecule that does not cross the blood–brain barrier (BBB) (green) and competes with RapaLink-1 (purple) for FKBP12 (yellow) binding in the periphery, protects mTORC1 activity outside of the CNS. Systemic co-administration of RapaLink-1 and RapaBlock allows for brain-specific inhibition of mTORC1.

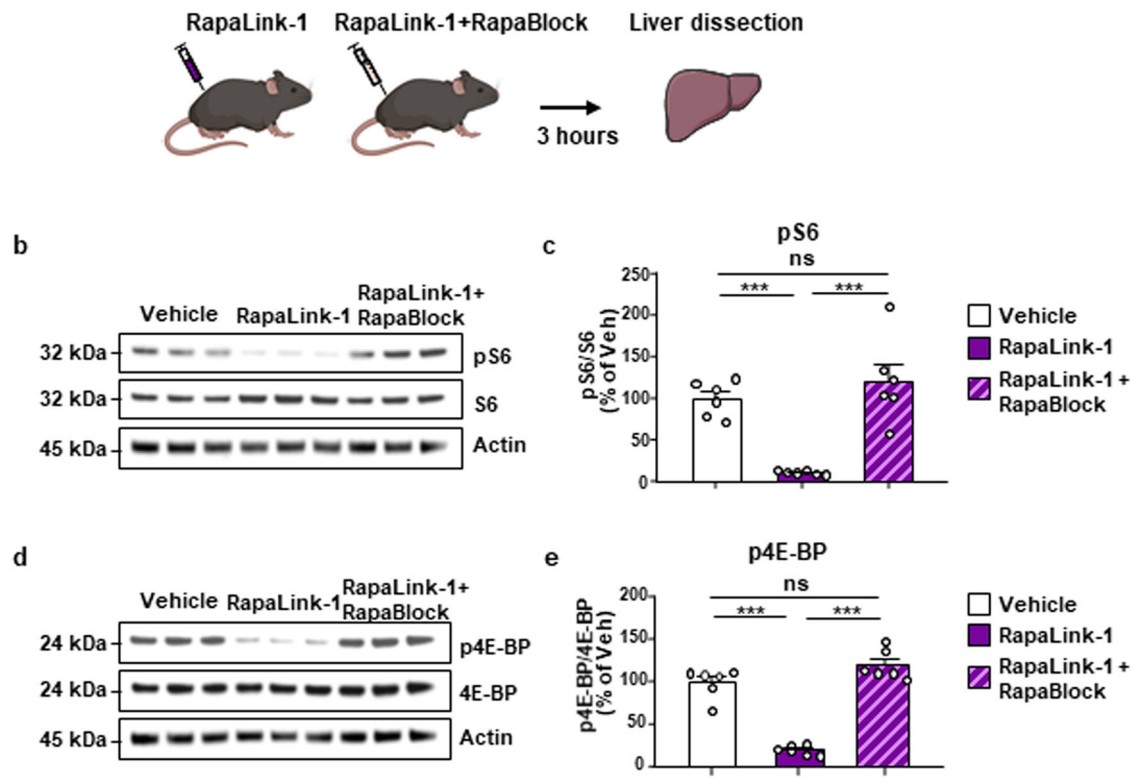

**Fig. 2 RapaBlock protects mTORC1 activity in the liver. a** Timeline of the experiment. Mice received a systemic administration of vehicle (white), RapaLink-1 alone (1 mg/kg, purple) or a combination of RapaLink-1 (1 mg/kg, purple) and RapaBlock (40 mg/kg, pink), and mTORC1 activity in the liver was measured 3 h later. **b, d** Representative images depict S6 phosphorylation (pS6) **b** and 4E-BP phosphorylation (p4E-BP) **d** (top panels), total protein levels of S6 **b** and 4E-BP **d** (middle panels), and actin **b, d** (bottom panels), which was used as a loading control. **c, e** Data are presented as the individual data points and mean densitometry values of the phosphorylated protein divided by the densitometry values of the total protein ± SEM and expressed as % of vehicle. Significance was determined using One-way ANOVA followed by Tukey's multiple comparisons test. RapaBlock protects mTORC1 activity in the liver. pS6 (One-way ANOVA: $F_{2,15} = 20.56$, $p < 0.0001$, $r^2 = 0.7327$; vehicle vs. RapaLink-1 $p = 0.0006$, vehicle vs. RapaLink-1+RapaBlock $p = 0.4796$, RapaLink-1 vs. RapaLink-1+RapaBlock $p < 0.0001$). p4E-BP (one-way ANOVA: $F_{2,15} = 42.77$, $p < 0.0001$, $r^2 = 0.9101$; vehicle vs. RapaLink-1 $p < 0.0001$, vehicle vs. RapaLink-1+RapaBlock $p = 0.0713$, RapaLink-1 vs. RapaLink-1+RapaBlock $p < 0.0001$). $n = 6$ per condition. ***$p < 0.001$, ns = non-significant.

activity in the periphery, mice received a systemic administration of RapaLink-1 alone (1 mg/kg) or a combination of RapaLink-1 (1 mg/kg) and RapaBlock (40 mg/kg) and mTORC1 activity in the periphery was measured 3 h later (Fig. 2a). The liver was chosen as a peripheral organ since the mTORC1 pathway has a critical role in hepatic function[7], and since chronic inhibition of mTOR in the liver has been implicated in liver toxicity[34]. As expected, RapaLink-1 administrated alone blocked the phosphorylation of the mTORC1 downstream targets, S6 (Figs. 2b, c) and 4E-BP (Fig. 2d, e) in the liver. In contrast, co-administration of RapaLink-1 and RapaBlock produced a complete protection of mTORC1 activity in the liver (Fig. 2b–e), demonstrating that RapaBlock protects mTORC1 activity in a peripheral organ.

**Rapablock protects against Rapalink-1-dependent side effects.**
As mentioned above, chronic administration of rapamycin produces in humans and rodents a broad range of undesirable effects such as body weight loss, impaired glucose metabolism[29,35], and liver toxicity[34]. We next examined whether RapaBlock could protect against these adverse effects caused by long-term mTORC1 inhibition in the periphery. Mice were chronically treated three times a week for 4 weeks with either RapaLink-1 alone (1 mg/kg) or with a combination of RapaLink-1 (1 mg/kg) and RapaBlock (40 mg/kg) (Fig. 3a). Similar to what was previously reported for rapamycin[35], chronic treatment of mice with RapaLink-1 led to a significant decrease in body weight (Fig. 3b).

However, the combination of RapaLink-1 and RapaBlock prevented the decrease in the weight of the mice (Fig. 3b).

Chronic inhibition of mTORC1 in the periphery has been linked to hyperglycemia and insulin resistance[35,36]. We therefore, determined whether long-term administration of RapaLink-1 causes glucose intolerance and whether RapaBlock blocks this effect. To do so, a fasting glucose tolerance test (GTT) was conducted during the fourth week of treatment of mice with RapaLink-1 (1 mg/kg) alone or RapaLink-1 (1 mg/kg) and RapaBlock (40 mg/kg). Blood glucose levels were markedly increased in mice chronically treated with RapaLink-1 (Fig. 3c). In contrast, blood glucose levels were similar in vehicle-treated vs. RapaLink-1+RapaBlock-treated mice, suggesting the RapaBlock also protects against the glucose intolerance side effect.

**Rapablock-1 protects against Rapalink-1-dependent liver toxicity.** Prolonged treatment with rapamycin in mice results in liver inflammation[34]. To determine whether chronic administration of RapaLink-1 produces a similar liver toxicity phenotype, the liver was dissected and harvested following four weeks of RapaLink-1 treatment, and liver inflammation was evaluated by measuring the phosphorylation level of the signal transducer and activator of transcription 3 (STAT3), a prerequisite for the activation of the transcription factor, as well as the expression of the fibrogenic markers, tissue inhibitor of metalloproteinase 1 (Timp1), and collagen alpha1(IV) (Col4a1)[34]. We found that

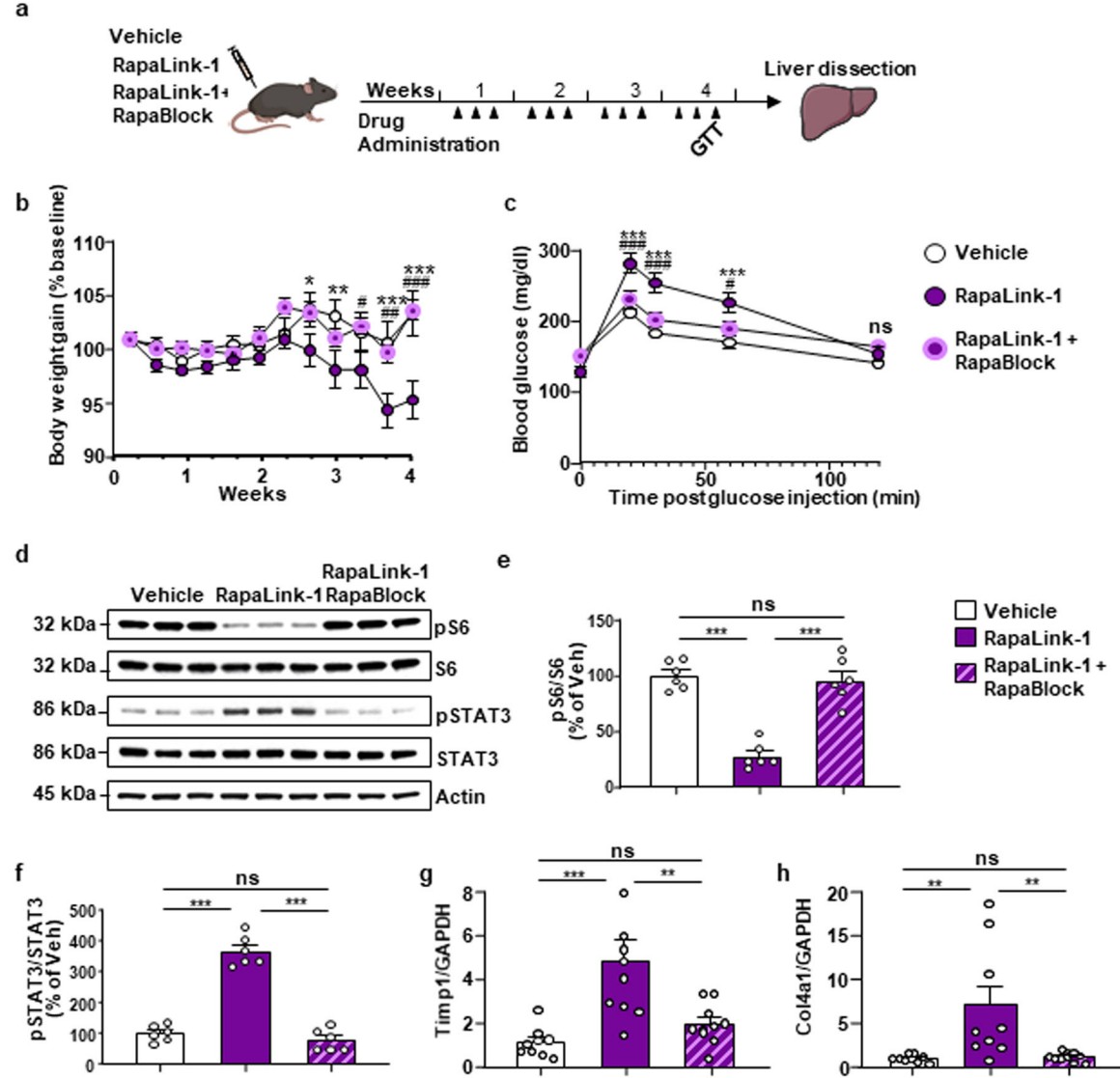

chronic administration of RapaLink-1 while inhibiting mTORC1 activity (Fig. 3d, e), robustly elevated the level of STAT3 phosphorylation at the Tyrosine 705 site (Fig. 3d, f), Four weeks of administration of RapaLink-1 also increased the mRNA levels of *Timp1* (Fig. 3g) and *Col4A1* (Fig. 3h), suggesting that the drug produces liver toxicity. Importantly, RapaBlock protected mTORC1 activity in the liver (Fig. 3d, e) and at the same time, RapaBlock prevented the increase in STAT3 phosphorylation (Fig. 3d, f), and the expression of liver toxicity markers (Fig. 3g, h), suggesting that RapaBlock eliminates liver toxicity issues associated with prolonged inhibition of mTORC1. In contrast, chronic administration of RapaBlock alone did not alter mTORC1 activity in the liver and did not change the level of toxicity markers (Supplementary Fig. 2). Together, these results suggest that mTORC1 inhibitors have limited utility due to significant adverse side effects such as reduction in body weight, glucose intolerance, and liver toxicity. However, the finding that RapaBlock is able to fully prevent these adverse effects, resulting from the sustained inhibition of mTORC1 in the periphery.

**Rapablock alone does not alter behavior.** To examine the utility of the approach for a CNS application, we first examined the behavioral consequences of chronic systemic administration of RapaBlock alone in mice. We reasoned that if RapaBlock does not cross the blood–brain barrier, it should produce no adverse cognitive effects when administered chronically. First, mice were treated with RapaBlock alone (40 mg/kg) for 6 weeks and a battery of behavioral tests were performed at the 3rd and 6th weeks-time point (Supplementary Fig. 3a). Chronic RapaBlock treatment did not alter sensorimotor coordination, as measured by the latency to fall from a rotarod apparatus (Supplementary Fig. 3b–d), anxiety-like behavior, as measured in an elevated plus-maze paradigm (Supplementary Fig. 3e–g), or recognition memory, which was tested using a novel object recognition paradigm (Supplementary Fig. 3h, i). These results suggest that RapaBlock has no behavioral effects on its own.

**Rapalink-1+Rapablock inhibit alcohol activation of mTORC1 in the NAc.** Alcohol activates mTORC1 in the NAc of rodents[18,19,23], and blockade of mTORC1 in the CNS attenuates numerous phenotypes associated with alcohol use including excessive alcohol intake[18,22,23]. Having demonstrated the ability of RapaBlock to protect the function of mTORC1 in the periphery and to block the adverse phenotypes stemming from chronic RapaLink-1 treatment without producing behavioral side effects, we examined whether this approach could produce a selective inhibition of mTORC1 activity in the brain and be a potentially effective treatment for AUD. To do so, we first

**Fig. 3 RapaBlock abolishes RapaLink-1-dependent weight loss, glucose intolerance, and liver toxicity. a** Timeline of experiment. Mice were treated three times a week with vehicle (white) RapaLink-1 alone (1 mg/kg, purple) or with a combination of RapaLink-1 (1 mg/kg, purple) and RapaBlock (40 mg/kg, pink) for 4 weeks, and body weight, glucose tolerance, and liver toxicity were evaluated. **b** Co-administration of RapaLink-1 and RapaBlock eliminates RapaLink-1-dependent body weight loss (Two-way ANOVA: effect of time ($F_{11,312} = 3.467$, $p = 0.0001$), the effect of treatment ($F_{2,312} = 27.55$, $p < 0.0001$) and interaction ($F_{22,312} = 1.731$, $p = 0.0233$); at day 8: vehicle vs. RapaLink-1 $p = 0.0440$, at day 9: vehicle vs RapaLink-1 $p = 0.0061$, at day 10: RapaLink-1 vs. RapaLink-1+RapaBlock $p = 0.0249$, at day 11: vehicle vs. RapaLink-1 $p = 0.0003$, RapaLink-1 vs. RapaLink-1+RapaBlock $p = 0.0014$, at day 12: vehicle vs. RapaLink-1 $p < 0.0001$, RapaLink-1 vs. RapaLink-1+RapaBlock $p < 0.0001$)). **c** Glucose tolerance test was performed during the last week of chronic drug treatment. Co-administration of RapaLink-1 and RapaBlock reduces RapaLink-1-dependent increase in blood glucose (two-way ANOVA: effect of time ($F_{11,312} = 3.467$, $p = 0.0001$), the effect of treatment ($F_{2,312} = 27.55$, $p < 0.0001$) and interaction ($F_{22,312} = 1.731$, $p = 0.0233$); at 20 min: vehicle vs. RapaLink-1 $p < 0.0001$, RapaLink-1 vs. RapaBlock $p = 0.0008$, at 30 min: vehicle vs. RapaLink-1 $p < 0.0001$, RapaLink-1 vs. RapaBlock $p = 0.0008$, at 60 min: vehicle vs. RapaLink-1 $p = 0.0003$, RapaLink-1 vs. RapaBlock $p = 0.0250$). **d–h** Co-administration of RapaLink-1 and RapaBlock protects against RapaLink-1-dependent liver toxicity. **d–f** The liver was dissected 24 h after the last drug administration and S6 and STAT3 phosphorylation were measured. **d** Representative images of pS6, total S6 (top panels), phospho-STAT3 (pSTAT), total STAT3 (middle panels), and actin (bottom panel). **e** Co-administration of RapaLink-1 and RapaBlock protects mTORC1 activity in the liver (One-way ANOVA: $F_{2,15} = 42.77$, $P < 0.0001$, $r^2 = 0.8508$; vehicle vs. RapaLink-1 $p < 0.0001$, vehicle vs. RapaLink-1+RapaBlock $p = 0.8715$, RapaLink-1 vs. RapaBlock $p < 0.0001$). **f** RapaBlock reverses RapaLink-1-dependent increase in STAT3 phosphorylation (one-way ANOVA: $F_{2,15} = 104.9$, $p < 0.0001$, $r^2 = 0.9333$; vehicle vs. RapaLink-1 $p < 0.0001$, vehicle vs. RapaLink-1+RapaBlock $p = 0.6087$, RapaLink-1 vs. RapaBlock $p < 0.0001$). **g**, **h** RapaBlock protects against RapaLink-1-dependent increase of fibrogenic markers, Timp1 (one-way ANOVA: $F_{2,25} = 13.18$, $p = 0.0001$, $r^2 = 0.3733$; vehicle vs. RapaLink-1 $p = 0.0004$, vehicle vs. RapaLink-1+RapaBlock $p = 0.5992$, RapaLink-1 vs. RapaLink-1+RapaBlock $p = 0.0041$), and Col4a1 (one-way ANOVA: $F_{2,25} = 7.446$, $p = 0.0029$, $r^2 = 0.4605$; vehicle vs. RapaLink-1 $p = 0.0069$, vehicle vs. RapaLink-1+RapaBlock $p = 0.9960$, RapaLink-1 vs. RapaLink-1+RapaBlock $p = 0.0069$). **b**, **c** Data are presented as mean ± SEM. Significance was determined using RM two-way ANOVA followed by Tukey's multiple comparisons test. Vehicle, $n = 9$, RapaLink-1, $n = 10$; RapaLink-1+RapaBlock, $n = 10$, * = RapaLink-1 vs. Vehicle, # = RapaLink-1 vs. RapaLink-1+RapaBlock. * or # $p < 0.05$, ** or ## $p < 0.01$ and *** or ### $p < 0.001$, ns = non-significant. **e**, **f** Data are presented as the individual data points and mean densitometry values of the phosphorylated protein divided by the densitometry values of the total protein ± SEM and expressed as % of vehicle. Significance was determined using One-way ANOVA followed by Tukey's multiple comparisons test. $n = 6$ per condition, ***$p < 0.001$, ns = non-significant. **g**, **h** Data are expressed as a ratio of TIMP1 or Col4a1 to total GAPDH and presented as individual data points and mean ± SEM. Significance was determined using one-way ANOVA followed by Tukey's multiple comparisons test. Vehicle, $n = 9$, RapaLink-1, $n = 9$, RapaLink-1+RapaBlock, $n = 10$, **$p < 0.01$, ***$p < 0.001$, ns = non-significant.

determined whether RapaLink-1 inhibits alcohol-dependent mTORC1 activity in the NAc. Male mice underwent 7 weeks of intermittent access to 20% alcohol in a two-bottle choice, a paradigm that models binge alcohol intake in humans[37]. Mice consuming water only for the same length of time were used as a control. On week 8, mice received a systemic administration of RapaLink-1 alone (1 mg/kg) or a combination of RapaLink-1 (1 mg/kg) and RapaBlock (40 mg/kg) 3 h before the last 24-hr drinking session and mTORC1 activity was measured in the NAc (Fig. 4a). Excessive alcohol intake increased phosphorylation of the mTORC1 targets S6 and 4E-BP in the NAc of mice consuming alcohol and treated with vehicle vs. animals consuming water only (Fig. 4b–e). Systemic administration of RapaLink-1 inhibited alcohol-dependent mTORC1 activation in the NAc in the absence or presence of RapaBlock (Fig. 4b–e). In contrast, systemic administration of RapaBlock alone had no effect on alcohol-dependent activation of mTORC1 in the NAc (Supplementary Fig. 4).

**Rapalink-1+Rapablock inhibit alcohol drinking and seeking**. Next, we tested if RapaLink-1 attenuates alcohol intake in the presence or absence of RapaBlock (Fig. 5a). RapaBlock administered alone had no effect on alcohol and water intake (Supplementary Fig. 5), and blood alcohol concentration (BAC) was also unaltered (Supplementary Fig. 6a, b). RapaLink-1 reduced alcohol intake and preference measured during the first 4 h of an alcohol drinking session (Fig. 5b, c), and at the end of a 24-hour session (Fig. 5e, f), Importantly, co-administration of RapaLink-1 and RapaBlock produced similar attenuation of alcohol intake and preference, compared with the vehicle group (Fig. 5b, c, e, f). Co-treatment of RapaLink-1 and RapaBlock did not affect water intake (Fig. 5d, g) or BAC (Supplementary Fig. 6c).

Finally, we tested whether the co-treatment of RapaLink-1 and RapaBlock attenuates alcohol operant self-administration. Mice underwent 7 weeks of intermittent access to 20% alcohol in a two-bottle choice paradigm and were then trained to press on an

active lever under a fixed ratio 2 (FR2) schedule to obtain a 20% alcohol reward (Fig. 6a, Supplementary Fig. 7). After establishing a stable baseline, with an average of 100 active lever presses and alcohol intake of 2.16 ± 0.17 g/kg/2 h (Supplementary Fig. 7), mice received a systemic administration of RapaLink-1 (1 mg/kg) and RapaBlock (40 mg/kg) 3 h before the operant self-administration session (Fig. 6a). Co-administration of RapaLink-1 and RapaBlock produced a robust reduction in alcohol self-administration as evidenced by the decrease in the number of total active but not inactive lever presses (Fig. 6b) as well as the cumulative (Fig. 6c) and the frequency (Fig. 6d) of lever presses. The reduction of lever presses resulted in a corresponding reduction in the alcohol intake (Fig. 6e). Next, to test whether RapaLink-1+RapaBlock drug treatment reduces alcohol-seeking, mice received a systemic administration of both drugs 3 h prior to an extinction session, during which presses of the active lever are not rewarded. The dual-drug treatment produced a significant decrease in the total number of lever presses (Fig. 6f) as well as the cumulative (Fig. 6g) and frequencies of lever presses (Fig. 6h), suggesting that RapaLink-1 reduces alcohol seeking and/or enhances extinction. Together, these data suggest that RapaLink-1 preserves its desirable inhibitory actions on mTORC1 in the brain when administered together with RapaBlock.

## Discussion
We show herein that RapaBlock provides full protection of mTORC1 activity in the periphery. The small molecule also prevents the detrimental side effects, resulting from chronic inhibition of the kinase in the periphery. We further present preclinical proof of concept data for the potential utility of the RapaLink-1+RapaBlock dual-drug administration strategy for the treatment of AUD.

Our data suggest that RapaBlock acts in the periphery but not in the CNS of mice. Specifically, RapaBlock blocked RapaLink-1's inhibitory actions on mTORC1 activity in the liver but did not affect mTORC1 activity in the brain. Administration of the small

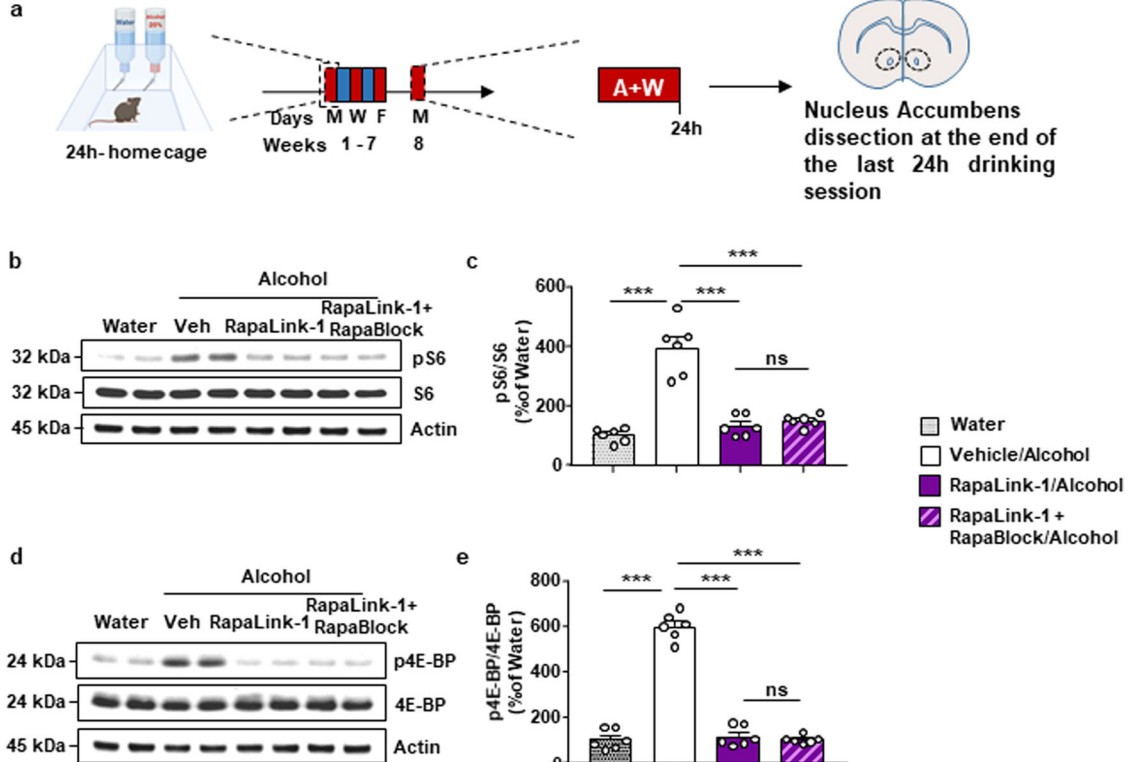

**Fig. 4 RapaLink-1 inhibits alcohol-dependent mTORC1 activation in the nucleus accumbens in the presence and absence of RapaBlock. a** Timeline of experiment. Mice underwent 7 weeks of IA20%2BC. On week 8, mice received a systemic administration of vehicle (white), RapaLink-1 alone (1 mg/kg, purple), or a combination of RapaLink-1 (1 mg/kg, purple) and RapaBlock (40 mg/kg, pink) 3 h before the beginning the drinking period, and the NAc was removed at the end of the last 24 h drinking session. **b, d** Representative images depict S6 phosphorylation (pS6) (**b**) and 4E-BP phosphorylation (p4E-BP) (**d**) (top panels), total protein levels of S6 (**b**) and 4E-BP (d) (middle panels), and actin (bottom panels). **c, e** RapaLink-1 inhibits alcohol-dependent mTORC1 activation activity in the presence and absence of RapaBlock pS6 (one-way ANOVA: $F_{3,20} = 41.58$, $p < 0.0001$, $r^2 = 0.8618$; vehicle vs. water $p < 0.0001$, vehicle vs. RapaLink-1 $p < 0.0001$, vehicle vs. RapaLink-1+RapaBlock $p < 0.0001$) and p4E-BP (one-way ANOVA: $F_{3,20} = 186$, $p < 0.0001$, $r^2 = 0.9654$; vehicle vs. water $p < 0.0001$, vehicle vs. RapaLink-1 $p < 0.0001$, vehicle vs. RapaLink-1+RapaBlock $p < 0.0001$). Data are presented as the individual data points and mean densitometry values of the phosphorylated protein divided by the densitometry values of the total protein ± SEM and expressed as % of vehicle. Significance was determined using one-way ANOVA Tukey's multiple comparisons test. $n = 6$ per condition. ***$p < 0.001$, ns = non-significant.

molecule by itself did not alter alcohol and water intake nor did it affect mice's locomotion, recognition memory, and anxiety-like behavior. Importantly, RapaBlock prevented numerous adverse side effects resulting from long-term inhibition of mTORC1 in the periphery. However, more studies are warranted to test RapaBlock's ability to prevent other major side effects produced in humans by mTORC1 inhibitors such as immunosuppression and diabetes[38]. Nevertheless, our data suggest that RapaBlock in combination with RapaLink-1 could potentially be used for CNS-specific applications such as AUD.

Furthermore, the approach described herein, enabling the separation between the desirable, CNS-mediated actions of a drug versus the undesirable periphery-mediated drug effects could in principle be used for the development of other CNS-targeted therapeutic approaches. For instance, Fyn kinase has been implicated in mechanisms underlying Alzheimer's disease[39], AUD[40], and opiate addiction[41], and small molecule inhibitors such as AZD0530 have been in development for the treatment of Alzheimer's disease[39]. Protecting the activity of the kinase in the periphery will enable the reduction of potential side effects and increase the safety of the inhibitor.

One caveat of our study is that it was conducted in only male mice. The decision to initially test the utility of the dual-drug approach in male mice was due to a previous report suggesting

that rapamycin does not reduce alcohol intake in female mice[42]. However, future studies are aimed to replicate Cozzoli et al.'s findings and also to determine the potential use of this strategy for other alcohol-dependent phenotypes in female mice. In addition, in the behavioral studies, RapaLink-1 was administered only once and further studies are needed to determine the beneficial effects of long-term treatment of rapalogs.

AUD is the third most preventable disease[3], unfortunately, drug development for the treatment of AUD has only been modestly successful to date[2]. Data obtained in rodents suggest that inhibition of mTORC1 in the brain dampens numerous adverse behaviors associated with alcohol use including excessive alcohol intake[18,22,23], habitual alcohol seeking[10], alcohol reward[18,22], reconsolidation of alcohol reward memories[24], and reinstatement of alcohol place preference[25]. In addition, and as shown herein, the dual-drug strategy is also beneficial for alcohol drinking phenotypes such as binge alcohol intake, alcohol operant self-administration, and alcohol-seeking and/or extinction. In contrast, inhibition of mTORC1 does not alter the consumption of natural rewarding substances, suggesting that inhibition of mTORC1 does not affect reward per se[10,18,22,24]. Furthermore, treatment of rodents with the mTORC1 inhibitor rapamycin does not cause aversion or reward, nor does it alter locomotion[18,24]. Putting together these preclinical rodent studies together with the

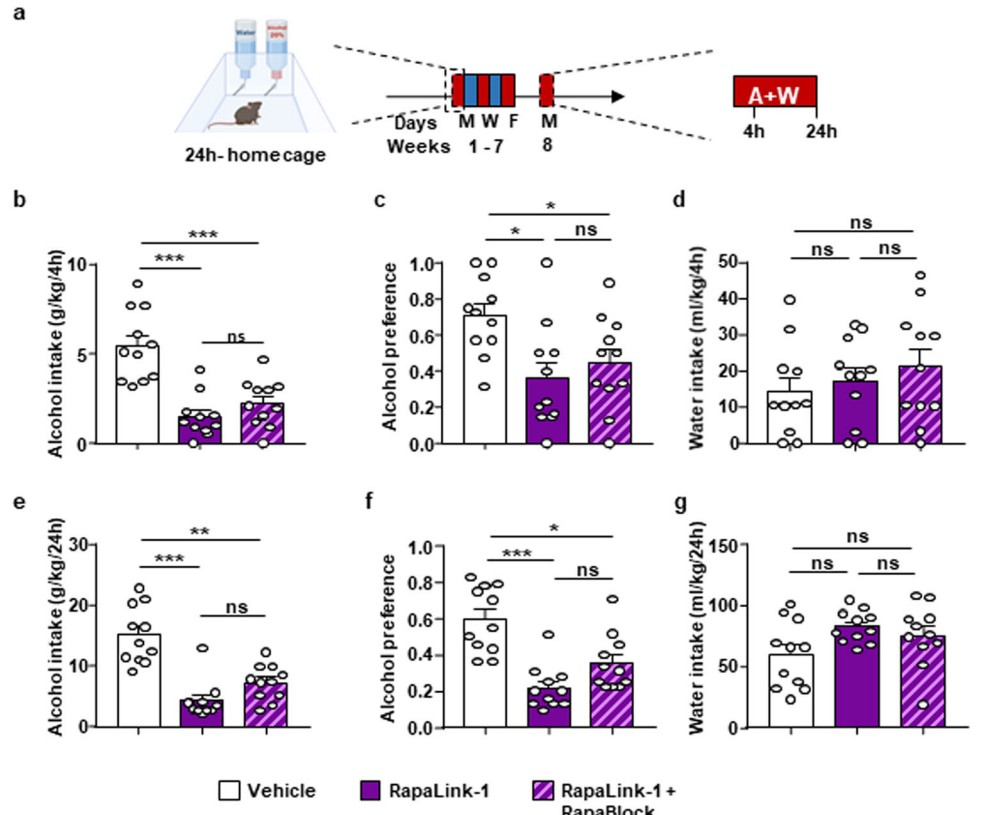

**Fig. 5 RapaLink-1 reduces alcohol intake and preference in the presence and absence of RapaBlock. a** Timeline of experiment. Mice underwent 7 weeks of IA20%2BC. On week 8, mice received a systemic administration of vehicle (white) RapaLink-1 alone (1 mg/kg, purple) or a combination of RapaLink-1(1 mg/kg, purple) and RapaBlock (40 mg/kg, pink) 3 h before the beginning of drinking session. Alcohol and water intake were measured 4 h (**b, d**) and 24 h later (**e, g**). Alcohol preference was calculated as the ratio of alcohol intake to total fluid intake at the end of the 4 h (**c**) and 24 h (**f**) drinking session. **b, c** Administration of RapaLink-1 alone or RapaLink-1+RapaBlock significantly decreased alcohol intake at the end of a 4 h session (one-way ANOVA: $F_{2,30} = 20.46$, $p < 0.0001$, $r^2 = 0.577$; vehicle vs. RapaLink-1 $p < 0.0001$, vehicle vs. RapaLink-1+RapaBlock $p < 0.0001$, RapaLink-1 vs. RapaLink-1+RapaBlock $p = 0.4820$), and alcohol preference (one-way ANOVA: $F_{2,30} = 5.482$, $p = 0.0094$, $r^2 = 0.2274$; vehicle vs. RapaLink-1 $p = 0.0102$, vehicle vs. RapaLink-1 +RapaBlock $p = 0.0462$, RapaLink-1 vs. RapaLink-1+RapaBlock $p = 0.4351$). **d** Water intake at the end of a 4-hour session was not affected by treatments (one-way ANOVA: $F_{2,30} = 0.7708$, $p = 0.4716$, $r^2 = 0.04887$). **e, f** Administration of RapaLink-1 alone or RapaLink-1+RapaBlock significantly decreased alcohol intake (one-way ANOVA: $F_{2,30} = 25.47$, $p < 0.0001$, $r^2 = 0.6294$; vehicle vs. RapaLink-1 $p < 0.0001$, vehicle vs. RapaLink-1+RapaBlock $p < 0.0001$, RapaLink-1 vs. RapaLink-1+RapaBlock $P = 0.1538$) and alcohol preference (one-way ANOVA: $F_{2,30} = 16.73$, $p < 0.0001$, $r^2 = 0.5272$; vehicle vs. RapaLink-1 $p < 0.0001$, vehicle vs. RapaLink-1+RapaBlock $p = 0.0029$, RapaLink-1 vs. RapaLink-1+RapaBlock $p = 0.1104$) at the end of a 24-hour session. **g** Water intake at the end of a 24-hour session was not affected by treatments (one-way ANOVA: $F_{2,30} = 2.966$, $p = 0.0668$, $r^2 = 0.1651$). Data are presented as individual data points and mean ± SEM. Significance was determined using one-way ANOVA Tukey's multiple comparisons test. $n = 11$ per condition. *$p < 0.05$, **$p < 0.01$, ***$p < 0.001$ and ns = non-significant.

data presented herein suggests that the Rapalink-1+RapaBlock dual-drug strategy may potentially be used in humans.

Interestingly, mTORC1 has been linked to neuroadaptations associated with numerous drugs of abuse[16]. For instance, rapamycin administration was shown to inhibit reconsolidation of cocaine and morphine reward memory as well as the reinstatement of cocaine self-administration[16]. These findings raise an attractive possibility that the binary drug strategy could be developed as a therapeutic option not only for AUD but also for the treatment of addiction to other drugs of abuse. Furthermore, treatment of rodents with rapamycin was reported to inhibit consolidation and reconsolidation of fear memory[43,44], thus this strategy may also be potentially useful for the treatment of post-traumatic stress disorder.

## Methods

**Animals**. Male C57BL/6 J mice (Jackson Laboratory, Bar Harbor, ME) were 6–7 weeks old at the beginning of the experiment. Mice were individually housed in a separate temperature- and humidity-controlled rooms (temperature and humidity were kept constant at 22 ± 2°C, and relative humidity was maintained at

50 ± 5%) under a 12-hour light/dark cycle (lights on at 07:00 AM) or a reversed 12 h light/dark cycle (lights on at 10:00 PM) with food and water available ad libitum.

**Reagents**. Anti-phospho-S6 (S235/236, 1:500), anti-S6 (1:1000), anti-phospho-4E-BP (T37/46, 1:500), anti-4E-BP (1:1000), anti-phospho-STAT3 (Y705, 1:500) and anti-STAT3 (1:500) antibodies were purchased from Cell Signaling Technology (Danvers, MA). Anti-Actin (1:10,000) antibodies, phosphatase Inhibitor Cocktails 2 and 3, and dimethyl sulfoxide (DMSO) were purchased from Sigma Aldrich (St. Louis, MO). The nitrocellulose membrane was purchased from EMD Millipore (Billerica, MA, USA). Enhanced chemiluminescence (ECL) was purchased from GE Healthcare (Pittsburg, PA). Donkey anti-rabbit and donkey anti-mouse horseradish peroxidase (HRP) were purchased from Jackson ImmunoResearch (West Grove, PA). AMV reverse transcriptase was purchased from Promega (Madison, WI). SYBR Green PCR Master mix was purchased from Thermo Fisher Scientific, Inc. (Waltham, MA, USA). EDTA-free complete mini Protease Inhibitor Cocktail was purchased from Roche (Indianapolis, IN). NuPAGE Bis-Tris precast gels and Phosphate buffered saline (PBS) were purchased from Life Technologies (Grand Island, NY). Bicinchoninic acid (BCA) protein assay kit was obtained from Thermo Scientific (Rockford, IL). ProSignal Blotting Film was purchased from Genesee Scientific (El Cajon, CA). Ethyl alcohol (190 proof) was purchased from VWR (Radnor, PA).

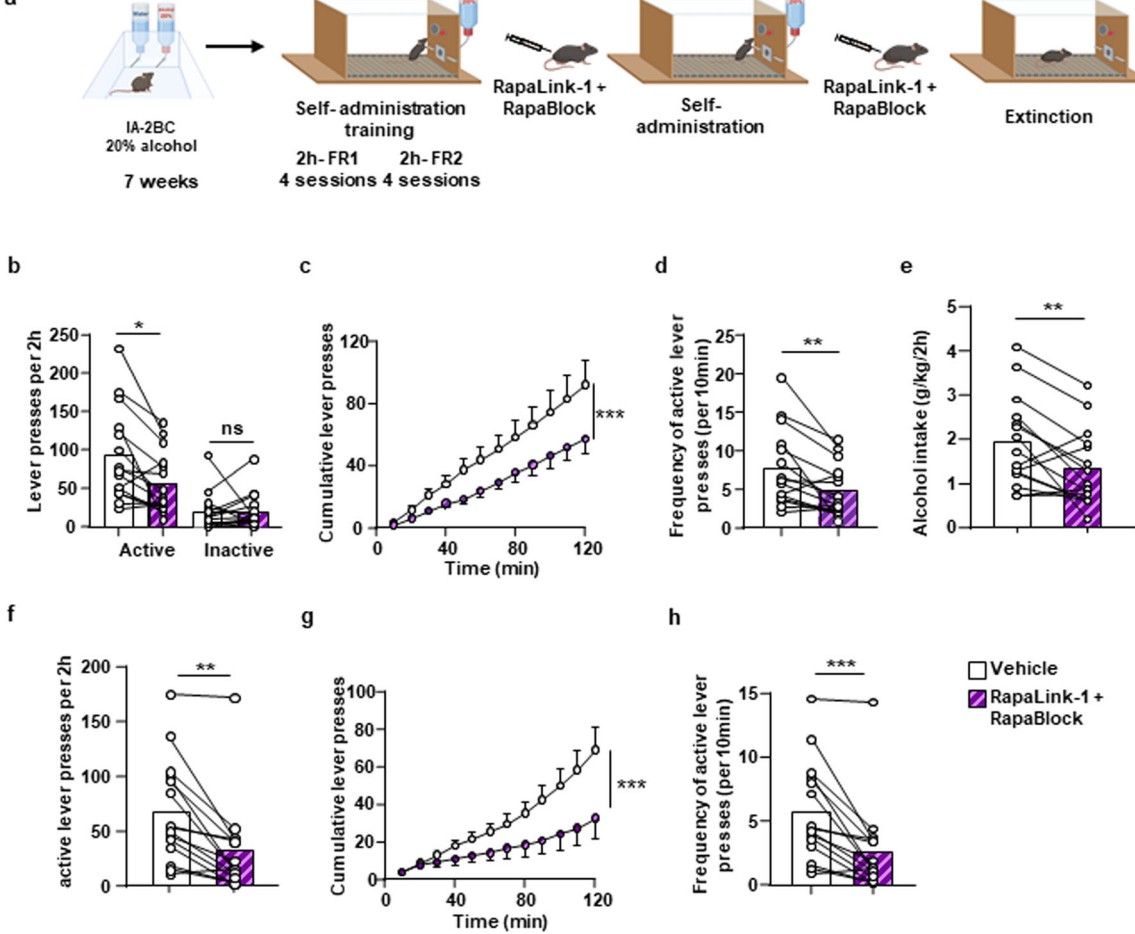

**Fig. 6 RapaLink-1 reduces alcohol self-administration and seeking in the presence of RapaBlock. a** Timeline of experiment. Mice that underwent 7 weeks of IA20%2BC were trained to self-administer 20% alcohol in operant chambers. After reaching a stable baseline of responding (Supplementary Figure 7), mice received a systemic administration of vehicle (white) or a combination of RapaLink-1(1 mg/kg, purple) and RapaBlock (40 mg/kg, pink) 3 h before the beginning of a self-administration session or before a single 2-hour extinction session. **b–e** Self-administration: the total number of lever presses (**b**), the cumulative number of lever presses (**c**), frequency of lever presses (**d**), and alcohol intake (**e**) were recorded. Administration of RapaLink-1+RapaBlock significantly decreased the total number of active lever presses (two-tailed paired $t$ test: $t = 3.329$, $p = 0.005$, $r^2 = 0.4418$), the cumulative number of lever presses (two-way ANOVA: main effect of treatment $F_{11,336} = 35.67$, $p < 0.0001$), the frequency of lever presses (two-tailed paired $t$ test: $t = 3.329$, $p = 0.005$, $r^2 = 0.4418$), and alcohol intake (two-tailed paired $t$ test: $t = 3.246$, $p = 0.0059$, $r^2 = 0.4294$). **f–h** Extinction: the total number of lever presses (**f**), cumulative number of lever presses (**g**), and frequency of active lever presses (**h**) were recorded. Administration of RapaLink-1+RapaBlock significantly decreased the total number of active lever presses (two-tailed paired $t$ test: $t = 3.336$, $p = 0.0024$, $r^2 = 0.2844$), the cumulative number of lever presses (two-way ANOVA: main effect of treatment $F_{11,336} = 33.63$, $p < 0.0001$), and the frequency of lever presses (two-tailed paired $t$ test: $t = 4.633$, $p = 0.0004$, $r^2 = 0.6053$) during an extinction session. Data are presented as individual data points and mean ± SEM. Significance was determined using a two-tailed paired $t$ test. n = 15 per condition. *$p < 0.05$, **$p < 0.01$, ***$p < 0.001$ and ns = non-significant.

**Rapalink-1 and Rapablock information**. RapaLink-1[21] and RapaBlock[32] (Supplementary Fig. 1) were synthesized in-house. Rapalink-1 was used at a concentration of 1 mg/kg, a dose that was found in our previous work to be sufficient for mTORC1 inhibition without causing toxicity[45]. This dose was previously used in behavioral studies[22]. RapaBlock was used at a concentration of 40 mg/kg, as this concentration is close to the solubility limit of the drug in its current formulation[32].

**Tissue harvesting**. Animals were killed and the brain and liver were rapidly removed an anodized aluminum block on ice. The NAc was isolated from a 1 mm thick coronal section located between +1.7 mm and +0.7 mm anterior to bregma according to the Franklin and Paxinos stereotaxic atlas (3rd edition). Collected tissues were immediately homogenized in 300 μl radioimmuno precipitation assay buffer containing (in mM: 50 Tris-HCl, pH 7.6, 150 NaCl, 2 EDTA), and 1% NP-40, 0.1% SDS and 0.5% sodium deoxycholate and protease and phosphatase inhibitor cocktails. Samples were homogenized by a sonic dismembrator. Protein content was determined using a BCA kit.

**Western blot analysis**. Equal amounts of homogenates from individual mice (30 μg) were resolved on NuPAGE Bis-Tris gels and transferred onto nitrocellulose membranes. Blots were blocked in 5% milk-PBS, 0.1% Tween 20 for 30 min and then incubated overnight at 4 °C with anti-pS6, anti-p4E-BP, and anti-pSTAT3 antibodies. Membranes were then washed and incubated with HRP-conjugated secondary antibodies for 2 h at room temperature. Bands were visualized using ECL. Membranes were then incubated for 30 min at room temperature in a stripping buffer containing 25 mM glycine-HCl and 1% (w/v) SDS, pH 3.0, and reprobed with anti-S6, anti-4E-BP, anti-STAT3, and anti-actin antibodies followed by secondary antibodies as described above. The optical density of the relevant band was quantified using ImageJ 1.44c software (NIH). Antibodies details are listed in Supplementary Table 1.

**cDNA synthesis and quantitative real-time PCR**. Total RNA extracted from liver samples were treated with DNase I. Synthesis of cDNA was performed using the AMV reverse transcriptase according to the manufacturer's instructions. The resulting cDNA was used for quantitative real-time PCR, using SYBR Green PCR Master mix. Thermal cycling was performed on QuantStudio 5 real-time PCR System (Thermo Fisher Scientific Inc.) using a relative calibration curve. The quantity of each mRNA transcript was measured and expressed relative to Glyceraldehyde-3-Phosphate dehydrogenase (GAPDH). Timp1[46], Col4a1[47], and GAPDH primers are listed in Supplementary Table 2.

**Preparation of solutions**. Alcohol solution was prepared from absolute anhydrous alcohol (190 proof) diluted to 20% alcohol (v/v) in tap water. RapaLink-1 (1 mg/kg)[22], and RapaBlock (40 mg/kg)[32] were dissolved in 5% DMSO, 5% Tween 80, 5% PEG300 and 85% saline. Vehicle contained 5% DMSO, 5% Tween 80, 5% PEG300, and 85% saline.

**Glucose tolerance test**. A GTT was performed as described previously[48]. In brief, mice were deprived of food for 6 h and then injected intraperitoneally (i.p.) with 1 g/kg glucose. Blood samples were taken from a tail vein nick at different time intervals (0, 15, 30, 60, and 120 min post glucose administration), and blood glucose level was analyzed using a Bayer Contour blood glucose meter and test strips.

**Behavioral testing**

*RapaBlock testing. Rotarod test*: Rotarod test was conducted as described previously[49]. Specifically, sensorimotor performance was assessed by the accelerating Rotarod apparatus (Rotarod 7650, Jones & Roberts). Each trial started at 4 rpm and reached 40 rpm speed after 300 s. Mice underwent three trials, with 5-min rest time in between trails. The trial ended when the mouse fell off the rod or completed one full revolution on the rod or when the speed of the apparatus reached 40 rpm (300 s). Latency to fall was scored in seconds, with 300 s as the maximum value.

*Novel object recognition (NOR) test*: The paradigm was conducted as described previously[50], with small modifications. Mice were first acclimated to the experimental room for 60 min. Afterward, two identical objects were placed in an open field (5 cm away from the walls), and mice were allowed to familiarize with both objects until they reached the criteria of 20 s of total exploration time. Six hours after the familiarization session, one familiar object was replaced by a novel object (the position of the novel object, left or right, was randomized between mice and groups). The mice were allowed to freely explore the open field until reaching the 20 s criterion of total exploration time. Exploration was characterized by the nose of the mouse directed toward an object at less than 2 cm of distance. Exploration time was recorded using the Ethovision XT video-tracking system (Noldus, Leesburg, VA, USA). The open field and the objects were cleaned with 75% ethanol and dried between each mouse and session.

*Elevated plus maze (EPM) paradigm*: The EPM paradigm was conducted as described previously[49]. Specifically, the EPM apparatus consists of two open and two closed arms (30 × 5 centimeters) with walls of 15 centimeters high and is elevated 40 centimeters above the ground. The arms extend from a central platform (5 × 5 centimeters) forming a plus sign. EPM testing took place in a quiet, dimly illuminated room. Each mouse was tested for 5 min after being placed in the center platform facing an open arm. The time spent on the closed arms and open arms of the EPM was scored. Arm entries were scored when an animal put all four paws into the arm. At the end of the test, the time spent in the open arms was expressed in seconds, and the total distance traveled in centimeters. Data were recorded via Ethovision XT video-tracking system (Noldus, Leesburg, VA, USA). EPM apparatus was cleaned with 75% ethanol and dried between animals and sessions.

*Drug administration*: Mice were systemically administered i.p. with vehicle or RapaBlock (40 mg/kg) on Mondays, Wednesdays, and Fridays and tested after 3 weeks and 6 weeks of chronic drug treatment.

**Rapablock and Rapalink-1 testing**

*Alcohol drinking paradigms. Two-bottle choice paradigm*: Mice underwent 7 weeks of intermittent access to 20% (v/v) alcohol in a two-bottle choice drinking paradigm (IA20%2BC) as described previously[51]. Specifically, mice had 24-hour access to one bottle of 20% alcohol and one bottle of water on Mondays, Wednesdays, and Fridays, with alcohol drinking sessions starting 2 h into the dark cycle. During the 24 or 48 h (weekend) of alcohol deprivation periods, mice had access to a bottle of water. The placement (right or left) of the bottles was alternated in each session to control for side preference. Two bottles containing water and alcohol in an empty cage were used to evaluate the spillage. Alcohol and water intake were measured at the 4-hr and 24-hr time points. The alcohol preference ratio was calculated as the volume of alcohol intake/total volume of fluid intake (water + alcohol).

*Drug administration*: Week 8 of the IA20%2BC, mice were administered i.p. with vehicle (5% DMSO, 5% Tween 80, 5% PEG300, and 85% saline), RapaLink-1 (1 mg/kg), RapaBlock (40 mg/kg) or a combination of RapaLink-1 (1 mg/kg) and RapaBlock (40 mg/kg) 3 h prior to the beginning of the drinking session, and alcohol and water intake were measured.

*Alcohol operant self-administration*: Operant self-administration of alcohol was conducted as described in ref. [52]. Specifically, prior to the operant self-administration training, mice were first subjected to IA20%2BC for 7 weeks as described above. Mice that drank more than 10 g/kg/24 h in the last week of IA20%2BC were selected for the alcohol operant self-administration training. Prior to the beginning of the training, each animal was handled for 1 min per day for three consecutive days. Self-administration training was conducted during the dark phase of the reversed dark/light cycle in operant chambers (length: 22 cm, width: 20 cm, height: 14 cm) equipped with two levers (1.5 cm in length, 11 cm apart, 2.5 cm from the grid floor) mounted at the opposite ends of the same wall (Med-Associates; Georgia, VT). The operant chambers included a reward port centered between the levers (0.5 cm from the grid floor) with photo-beams to allow monitoring of reward magazine visits, a light centered above the reward magazine,

and a tone-delivering tweeter situated on the opposite wall of the levers. Each chamber was housed within a sound-attenuating box with a fan providing background noise and ventilation. Mice underwent five daily self-administration sessions a week (Mon to Fri). Sessions started with the presentation of the 2 levers: responding on the active lever resulted in the delivery of a 20% alcohol via a motorized dipper that held 10 ml of liquid in the magazine. Reward delivery was paired with a 3 s tone (2900 Hz) and the illumination of the cue-light above the magazine. Responding to the other lever, i.e., the inactive lever, had no programmed consequences. The number and timing of the active and inactive lever presses, reward port visits, and reward deliveries were recorded during each session (Supplementary Fig. 7). Self-administration training was initiated under a fixed ratio (FR) 1, i.e., one lever press results in the delivery of one reward, for 3 6-hours per day sessions (10:00–16:00) followed by three 4 h per day sessions (11:00–15:00). Afterward, the sessions lasted for 2 h per day (13:00–15:00). Four sessions under FR1 schedule followed by four FR2 sessions were conducted.

*Drug administration*: Mice were systemically administered with vehicle or RapaLink-1 (1 mg/kg) + RapaBlock (40 mg/kg), 3 h prior to the beginning of an operant self-administration session or before a 2-hr extinction session (i.e., lever presses were not paired with alcohol deliveries). All drug testing was performed using a "within-subject" design in which mice received both treatments in counterbalanced order, with at least four standard FR2 sessions between tests.

**BAC measurement**. The BAC procedure was conducted as described in ref. [18] with modifications. Mice were systemically co-administered with RapaBlock (40 mg/kg) or RapaLink-1 (1 mg/kg) + RapaBlock (40 mg/kg) or vehicle. Three hours after drug administration, mice received an i.p. injection of 2 g/kg of alcohol, and blood was collected intracardially in heparinized capillary tubes 30 min later. Serum was extracted with 3.4% trichloroacetic acid followed by 5-minute centrifugation at $420 \times g$ and assayed for alcohol content using the NAD-NADH enzyme spectrophotometric method[53]. BAC was determined by using a standard calibration curve.

**Statistical analysis**. GraphPad Prism 7.0 (GraphPad Software Inc., La Jolla, CA) was used to plot and analyze the data. D'Agostino–Pearson normality test and F-test/Levene tests were used to verify the normal distribution of variables and the homogeneity of variance, respectively. Data were analyzed using the appropriate statistical test, including two-tailed paired $t$ test, two-tailed unpaired $t$ test, one-way analysis of variance (ANOVA), and two-way ANOVA followed by post hoc tests as detailed in Figure Legends. All data are expressed as mean ± SEM, and statistical significance was set at $p < 0.05$.

**Reporting summary**. Further information on research design is available in the Nature Research Reporting Summary linked to this article.

## Data availability
Raw biochemical data are available in the Supplementary Information section. Source data are provided with this paper.

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

## Acknowledgements

The authors thank Dr. Jeffrey Moffat for his input. This research was supported by the National Institute of Alcohol Abuse and Alcoholism, R01 AA027474 (D.R.), Damon Ruyon Cancer Research Foundation (DRG-2281-17) (Z.Z.), National Cancer Institute R01CA221969 (K.M.S.), the Michael J. Fox Foundation P0536220 (K.M.S.), the Samuel Waxman Research Foundation (K.M.S.) and the Howard Hughes Medical Institute (K.M.S.). Cartoons depicted in the Figures were created using BioRender.

## Author contributions

Y.E. designed the study, conducted the in vivo experiments, analyzed the data, and wrote the manuscript, Z.Z. designed and synthesized the chemical compounds, K.P. conducted the biochemical experiments and analyzed the data, D.S. participated in the behavioral experiments, K.M.S. designed the chemical compounds, D.R. designed the study and wrote the manuscript.

## Competing interests

K.M.S. and Z.Z. are co-inventors on patent applications covering RapaBlock owned by UCSF. K.M.S. is an inventor on patents covering RapaLink-1 owned by UCSF and licensed to Revolution Medicines. K.M.S. receives monetary and stock compensation and is a co-founder and SAB member of Revolution Medicines. All other authors do not have financial or non-financial competing interests.
