## [Peer Review File · Nature Communications]

Reviewer #1 (Remarks to the Author):

This is an exceptionally solid, exciting and with a few minor exceptions listed under "minor" very well written paper. It builds on major prior advances made by one of the leading groups in the alcohol field, in which this group has previously shown that inhibition of mTORC1 activity has beneficial effects for a spectrum of alcohol-addiction related behaviors.

Here, they show that these beneficial effects can be obtained using a brain penetrant mTORC1 inhibitor while protecting peripheral organs against know toxicity of the prototypical mTORC1 inhibitor rapamycin and its analogs using a small molecule drug that prevents binding of the inhibitor to a chaperone protein that is essential for its activity.

The paper is conceptually straightforward in the best sense of the word. The data are very convincing. This is a very important contribution that definitely should be published.

In Figs 2 and 3, the purist view would normally be that demonstrating preventions of Rapalink effects by RapaBlock should be done using a full 2 x 2 factorial design, where the prevention is properly tested by the interaction term. As it is, the RapaBlock-only group is missing, so that can't be done. This is, however, one of the rare occasions on which the purist approach really isn't all that important, since the authors independently examined the lack of tox from RapaBlock.

The prevention of kinase inhibition in the liver by RapaBlock (Fig 2) is extremely impressive. My understanding this is measured 3h after single or co-administration. From a drug development standpoint, I would like to know a little bit more about the pharmacokinetics of the two drugs – what are the respective half-lives? Is all the inhibitor gone by the time RapaBlock has been eliminated? What are the respective routes of elimination? The chronic experiment goes a long way to convincing me that the protective strategy works, but again, from a medication development perspective, it would be important to know how sensitive this strategy would be to missed doses, drug-drug interactions that influence ADME etc.

It is very nice that the authors carefully examined potential behavioral / cognitive / affective consequences of chronic RapaBlock administration alone.

Minor / editorial

Intro

It is easy to agree with the authors on the importance of identifying novel mechanisms that can be targeted for treatment of alcoholism, but in their eagerness to argue for the rationale for this, the authors go a bit over the top in the first paragraph. The rise in prevalence is probably driven by altered diagnostic criteria / procedures. Currently approved modern medications, naltrexone and acamprosate, are not so bad. The effect size of NTX is not lower than many other treatments in clinical medicine, and the safety and tolerability profile is very good. The greatest problem with these medications is that they are not used sufficiently. In that context, portraying them as being ineffective and fraught with adverse effects is neither correct, nor helpful. We simply need more treatments.

“Rapamycin” – not a proper name, should be lower case

“Mechanistic target of Rapamycin complex 1 (mTORC1) represents an ideal drug target for the treatment of AUD” – again, a bit over the top. It is a very exciting target, but ideal? From a drug development standpoint, an ideal “druggable” target is a GPCR or ligand gated channel. Intracellular enzymes are traditionally not considered ideal drug targets, in particular outside the immunology and cancer fields. This is not to say that I don’t share the author’s enthusiasm.

“suppression thrombocytopenia” – this is not a term. I assume authors refer to bone marrow suppression resulting in thrombocytopenia

“hyperlipodemia” – correct spelling

Conclusions

“Our data suggest that RapaBlock while acting in the periphery, the drug is inert in the CNS.” – the sentence is not well put together, please correct

“AUD is the third most preventable disease” – again, in need to rephrasing. “most preventable” doesn’t make sense. Most common?

Reviewer #2 (Remarks to the Author):

The manuscript entitled 'Brain-Specific Inhibition of mTORC1 by a Dual Drug Strategy: A Novel Approach for the Treatment of Alcohol Use Disorder' by Ehinger, Zhang and colleagues, reports on a series of experiments aimed at testing the potential usefulness of a binary pharmacology approach to treat alcohol use disorders. The authors tested the effects of a new small molecule drug, RapaBlock, in combination with the specific mTORC1 inhibitor RapaLink-1 to reduce alcohol intake and preference while avoiding (some of) the undesired side-effects of peripheral mTORC1 inhibition in male mice. The RapaBlock+RapaLink-1 combination successfully attenuated alcohol-induced S6 phosphorylation in the nucleus accumbens, reduced but did not affect pS6 or p4E-BP levels in the liver (this last effect was evident when the drugs were injected either acutely or chronically). Moreover, fibrogenic markers in the liver were increased by mTORC1 inhibition alone but remained unaffected when RapaBlock was co-administered. A similar effect was observed for the levels of pSTAT3 in the liver, used as a proxy for liver inflammation.

This is a very interesting study with great translational potential. The experiments have been adequately executed and the results are clear and sound, with important implications for the treatment of alcohol-related disorders. Indeed, the potential use of this new molecule may be an important step into more effective treatments for alcohol use disorders using mTORC1 inhibitors without the undesired side effects of these drugs. The results are highly novel, in fact, the design and general pharmacological effects of the molecule are not published yet (apart from a preprint deposited in bioRxiv, as indicated in the manuscript's reference list).

Although the work is robust, there are some issues that need to be addressed:

1.- What are the effects of Rapalink-1 with and without RapaBlock on pS6 and p4E-BP in other brain areas important for compulsive drug use such as the dorsolateral striatum? More brain areas should be tested to see if the effects reported are specific to the nucleus accumbens. Also, a justification of dose selected (presumably, based on the preprint describing RapaBlock) should be provided.

2.- The intervention proposed by the authors is focused on alcohol intake and preference, but other behavioural components of addictions may be examined such as habitual drug intake or seeking, relapse, or incubation of seeking. The authors have tested these other features in the past. In addition, it would be crucial to determine if the effects observed last for more than 24 hours. Is the reduction of alcohol intake and preference still observed once Rapalink-1+Rapablock treatment is withdrawn?

3.- Does this approach leave sucrose preference unaffected? Arguably, if these molecules are to be used in the future in humans, anhedonic effects should be avoided.

4.- Could RapaBlock affect long-term memory? The authors should test longer training-test intervals (such as 24 h) in the NOR test.

5.- What are the effects of RapaBlock alone in the liver? Are the effects observed in fibrogenic gene expression markers also observed if protein levels are tested? Where the liver enzymes affected?

6.- Does Rapablock alone modify alcohol intake or blood alcohol levels?

7.- Although the authors acknowledge this limitation, other peripheral organs, such as the bone marrow or pancreas, should be examined to rule gain a deeper understanding of the usefulness of this binary strategy to prevent undesired side effects of chronic mTORC1 inhibition.

8.- The statistical analyses are adequate but I would recommend that the authors report the effect sizes, using the appropriate statistic.

9.- The authors may want to acknowledge the caveat of the absence of experiments in females.

Minor comments:

1.- Figure 1 seems unnecessary to me.

2.- Fig. 4 C Legend: Does 'Shame' refer to VH?

3.- Figure 4 A Add 'last' between 'the' and '24 h' in the text below the brain slice diagram.

Reviewer #3 (Remarks to the Author):

This report from Ehinger and colleagues provides exciting findings relating to possible pharmacological therapies for alcohol use disorder (AUD). Prior work has shown in rodent models that mTORC1 Inhibitors, such as rapamycin, reduce neurobiological impacts of alcohol and alcohol intake itself. The author and her lab have spearheaded the bulk of this prior body of work. I am familiar with this work and find it strong and compelling. However, a major drawback has prevented the extension of this work to humans. And that is that rapamycin and other mTORC1 inhibitors have clear, major negative side effects in humans when used chronically. To address this, the authors

have taken an innovative approach in devising a way to block the disadvantageous peripheral effects, but not the desired central nervous system effects, of mTORC1 inhibitors by co-administering an agent that prevents peripheral action but does not cross the blood brain barrier, leaving central drug actions intact.

Through a series of animal control studies, the authors demonstrate protected liver function and glucose levels when the mTORC1 antagonist Rapalink1 is given with the blocking agent, 'rapablock', itself. This team has previously shown that Rapalink1 is an mTORC1 antagonist with similar or in some case improved attributes to rapamycin.

The authors go on to use molecular biology and behavioral pharmacology in mice to characterize the impacts of these two compounds. The authors show that alcohol intake is reduced in the standard animal model of high-drinking C57 mice. The experiments are well-designed, with the proper controls, the sample sizes are sufficient for me to be convinced of the findings. I am very pleased to report that I have no major issues with the work as presented. It is a timely and significant advance that moves us closer to treatment in humans.

There are some minor issues with the writing and presentation and a few questions, listed below.

1. The following sentence will lead some to believe there are human studies in this paper. Needs better wording: We show that the dual administration of RapaLink-1 with RapaBlock, abolishes RapaLink-1- dependent mTORC1 inhibition in the liver and blocks adverse side effects detected in humans including body weight loss, glucose intolerance and liver toxicity.
2. Can the authors comment on the likely use of this drug combination chronically. The rodent experiment reported findings from an acute injection. I am not suggesting this study be added to this work as it is already an extensive investigation.
3. Line 42, recommend change 'use' to 'intake'
4. Line 53, change semi-colon after widespread to a comma unless the wording of the following phrase is edited to become a clause.
5. Line 210: Needs rewording; Our data suggest that RapaBlock while acting in the periphery, the drug is inert in the CNS.
6. Must be clear to the reader that only males were tested; should be in abstract.

We thank the reviewers for the thoughtful and positive review of the manuscript. We conducted additional experiments and revised the manuscript carefully to address the questions and comments raised by the reviewers. We hope that the reviewers will find our revised manuscript acceptable for publication.

Below is a list of new experiments that we have added to the revision:

- To address Reviewer 2, point 3, we conducted an additional behavioral experiment and determined whether RapaLink-1 reduces operant alcohol self-administration and/or alters lever presses during an extinction session in the presence of RapaBlock (**Figure 6A**). As shown in new **Figure 6B-E**, administration of Rapalink-1 reduces alcohol lever presses and alcohol intake. As shown in **Figure 6F-H**, Rapalink-1+Rapablock also reduces presses of the lever previously associated with alcohol suggesting that mTORC1 inhibition attenuates seeking and/or enhances extinction.
- To address the Reviewer 2 point 7, we measured the level of mTORC1 activation, STAT3 phosphorylation and the expression of liver toxicity markers following chronic administration of Rapablock. As shown in new **Supplementary Figure 2**, RapaBlock does not alter mTORC1 activation nor does it produce liver toxicity.
- To address Reviewer 2 point 8, we examined whether Rapablock alters alcohol metabolism. As shown in new **Supplementary Figure 6A**, Rapablock does not alter blood alcohol concentration (BAC).

Reviewer #1

This is an exceptionally solid, exciting and with a few minor exceptions listed under "minor" very well written paper. It builds on major prior advances made by one of the leading groups in the alcohol field, in which this group has previously shown that inhibition of mTORC1 activity has beneficial effects for a spectrum of alcohol-addiction related behaviors.

Here, they show that these beneficial effects can be obtained using a brain penetrant mTORC1 inhibitor while protecting peripheral organs against know toxicity of the prototypical mTORC1 inhibitor rapamycin and its analogs using a small molecule drug that prevents binding of the inhibitor to a chaperone protein that is essential for its activity. The paper is conceptually straightforward in the best sense of the word. The data are very convincing. This is a very important contribution that definitely should be published. In Figs 2 and 3, the purist view would normally be that demonstrating preventions of RapaLink effects by RapaBlock should be done using a full 2 x 2 factorial design, where the prevention is properly tested by the interaction term. As it is, the RapaBlock-only group is missing, so that can't be done. This is, however, one of the rare occasions on which the purist approach really isn't all that important, since the authors independently examined the lack of tox from RapaBlock.

1. The prevention of kinase inhibition in the liver by RapaBlock (Fig 2) is extremely impressive. My understanding this is measured 3h after single or co-administration. From a drug development standpoint, I would like to know a little bit more about the pharmacokinetics of the two drugs – what are the respective half-lives? Is all the inhibitor gone by the time RapaBlock has been eliminated? What are the respective routes of elimination? The chronic experiment goes a long way to convincing me that the protective strategy works, but again, from a medication development perspective, it would be important to know how sensitive this strategy would be to missed doses, drug-drug interactions that influence ADME etc.

Answer: We are carrying out pharmacokinetics studies, but because of the complexity of having to extract both Rapalink-1 and Rapablock from the FKBP12 complex—and the potency of RapaLink-1 (sub nM), such studies require us to design and optimize new protocols, and these

turned out to be challenging. We will continue our efforts and publish the results in our subsequent work. We hope the reviewer understands.

It is very nice that the authors carefully examined potential behavioral / cognitive / affective consequences of chronic RapaBlock administration alone.

Minor / editorial

Intro

2. It is easy to agree with the authors on the importance of identifying novel mechanisms that can be targeted for treatment of alcoholism, but in their eagerness to argue for the rationale for this, the authors go a bit over the top in the first paragraph. The rise in prevalence is probably driven by altered diagnostic criteria / procedures. Currently approved modern medications, naltrexone and acamprosate, are not so bad. The effect size of NTX is not lower than many other treatments in clinical medicine, and the safety and tolerability profile is very good. The greatest problem with these medications is that they are not used sufficiently. In that context, portraying them as being ineffective and fraught with adverse effects is neither correct, nor helpful. We simply need more treatments.

Answer: We apologize for misstating the efficacy of current available medications, we toned down the text throughout the manuscript, and took out the sentence regarding the side effects of the current medications.

3. “Rapamycin” – not a proper name, should be lower case

Answer: Rapamycin is now in lower case, except for when we spelled out mTORC1.

4. “Mechanistic target of Rapamycin complex 1 (mTORC1) represents an ideal drug target for the treatment of AUD” – again, a bit over the top. It is a very exciting target, but ideal? From a drug development standpoint, an ideal “druggable” target is a GPCR or ligand gated channel. Intracellular enzymes are traditionally not considered ideal drug targets, in particular outside the immunology and cancer fields. This is not to say that I don’t share the author’s enthusiasm.

“suppression thrombocytopenia” – this is not a term. I assume authors refer to bone marrow suppression resulting in thrombocytopenia “hyperlipodemia” – correct spelling

Answer: The word “ideal” was replaced with “attractive”.

Conclusions

5. “Our data suggest that RapaBlock while acting in the periphery, the drug is inert in the CNS.” – the sentence is not well put together, please correct

Answer: The sentence has been corrected to “Our data suggest that RapaBlock acts in the periphery but not in the CNS of mice”.

6. “AUD is the third most preventable disease” – again, in need to rephrasing. “most preventable” doesn’t make sense. Most common?

Answer: The statement is based on information provided by NIAAA indicating that alcohol is third leading preventable cause of death in the United States.

<https://www.niaaa.nih.gov/publications/brochures-and-fact-sheets/alcohol-facts-and-statistics>.

Reviewer #2 (Remarks to the Author):

The manuscript entitled 'Brain-Specific Inhibition of mTORC1 by a Dual Drug Strategy: A Novel Approach for the Treatment of Alcohol Use Disorder' by Ehinger, Zhang and colleagues, reports on a series of experiments aimed at testing the potential usefulness of a binary pharmacology approach to treat alcohol use disorders. The authors tested the effects of a new small molecule drug, RapaBlock, in combination with the specific mTORC1 inhibitor RapaLink-1 to reduce alcohol intake and preference while avoiding (some of) the undesired side-effects of peripheral mTORC1 inhibition in male mice. The RapaBlock+RapaLink-1 combination successfully attenuated alcohol-induced S6 phosphorylation in the nucleus accumbens, reduced but did not affect pS6 or p4E-BP levels in the liver (this last effect was evident when the drugs were injected either acutely or chronically). Moreover, fibrogenic markers in the liver were increased by mTORC1 inhibition alone but remained unaffected when RapaBlock was co-administered. A similar effect was observed for the levels of pSTAT3 in the liver, used as a proxy for liver inflammation. This is a very interesting study with great translational potential. The experiments have been adequately executed and the results are clear and sound, with important implications for the treatment of alcohol-related disorders. Indeed, the potential use of this new molecule may be an important step into more effective treatments for alcohol use disorders using mTORC1 inhibitors without the undesired side effects of these drugs. The results are highly novel, in fact, the design and general pharmacological effects of the molecule are not published yet (apart from a preprint deposited in bioRxiv, as indicated in the manuscript's reference list). Although the work is robust, there are some issues that need to be addressed:

1. What are the effects of Rapalink-1 with and without RapaBlock on pS6 and p4E-BP in other brain areas important for compulsive drug use such as the dorsolateral striatum? More brain areas should be tested to see if the effects reported are specific to the nucleus accumbens.

Answer: We previously conducted an extensive survey of alcohol-dependent activation of mTORC1 in striatal regions of mice and rats¹. Using both western blot analysis and immunohistochemistry, we detected mTORC1 activation only in the nucleus accumbens but not in the dorsolateral striatum or the dorsomedial striatum of alcohol consuming rodents¹. We added the information to the introduction. Since mTORC1 is not activated by alcohol in other striatal regions, we do not believe that adding the suggested experiment will add new information to the current study.

2. Also, a justification of dose selected (presumably, based on the preprint describing RapaBlock) should be provided.

Answer: We chose to use 1 mg/kg RapaLink-1, a concentration found in our previous work to be sufficient for mTORC1 inhibition without causing toxicity², and we have successfully used this dose in for our alcohol behavioral studies³. We chose a concentration of 40 mg/kg for RapaBlock, as this concentration is close to the solubility limit of the drug in the current formulation (please also see⁴). We added the structure of both compounds to the manuscript (**Supplementary Figure 1**) and expand on dose justification in the methods.

3. The intervention proposed by the authors is focused on alcohol intake and preference, but other behavioural components of addictions may be examined such as habitual drug intake or seeking, relapse, or incubation of seeking. The authors have tested these other features in the past.

Answer: As we detailed in the introduction, we previously conducted a series of studies in mice and rats in which we found that in addition to the attenuation of alcohol intake^{3,5,6}, rapamycin and/or rapalink-1 block alcohol-place preference^{3,5} and thus reward, reconsolidation of alcohol self-administration in rats⁷, and reinstatement of alcohol place preference in mice⁸,

both are models of relapse, alcohol sensitization and habitual alcohol seeking⁹. We expanded on these findings in the introduction and discussion.

In addition, to address the reviewer's point and to gain more information on the utility of the multidrug approach, we conducted additional experiments and show that RapaLink-1+RapaBlock reduces mice operant self-administration (**Figure 6B-E**), as well as alcohol seeking (**Figure 6F-H**). We added the new findings to the manuscript.

4. In addition, it would be crucial to determine if the effects observed last for more than 24 hours. Is the reduction of alcohol intake and preference still observed once RapaLink-1+RapaBlock treatment is withdrawn?

Answer: To address the reviewer's comment, we examined whether RapaLink-1+RapaBlock-mediated reduction of alcohol drinking is long-lasting. We found that operant self-administration goes back to baseline 24 hours post drug treatment suggesting that the attenuation of alcohol self-administration is short-lasting (**Rebuttal Figure 1**). Having said that, Rapalink-1 produced long lasting effect when given prior to the first drink of alcohol³, and rapamycin erases reward seeking memory when given after a 2 weeks of extinction⁷. These data suggest that the length of alcohol exposure (first drink vs. several months) and the timing of drug treatment (after two week of extinction vs in between operant self-administration sessions) influence whether or not the disruption of alcohol-dependent neuroadaptations is long-lasting. In addition, at this point we do not know if repeated drug treatment will increase the length of the effect. These important questions will be examined in the future.

Rebuttal Figure 1. Rapalink-1+Rapablock effect on alcohol self-administration is abolished after 24 hours. (A) Number of active and inactive lever presses and (B) alcohol intake 24 hours after a single injection. RapaLink-1+RapaBlock had no effect on lever presses (Two tailed paired t test: $t = 1.412$, $P = 0.1799$, $r^2 = 0.1246$) and alcohol intake (Two tailed paired t test: $t = 1.818$, $P = 0.0905$, $r^2 = 0.1910$) 24 hours post drugs administration.

5. Does this approach leave sucrose preference unaffected? Arguably, if these molecules are to be used in the future in humans, anhedonic effects should be avoided.

Answer: The reviewer is correct to point this out. However, we do not believe that anhedonia is likely to be an issue for the following reasons: a. Sucrose consumption does not activate mTORC1 in the nucleus accumbens suggesting that the activation of mTORC1 is specific to alcohol and is not shared with other rewarding substances¹, b. RapaLink-1 does not alter saccharine intake³; c. Rapamycin does not alter reinstatement or reconsolidation of sucrose self-administration⁷; and d. Rapamycin does not alter habitual sucrose seeking⁹.

6. Could RapaBlock affect long-term memory? The authors should test longer training-test intervals (such as 24 h) in the NOR test.

Answer: According to our findings in this manuscript and in ⁴, RapaBlock does not cross the blood brain barrier. Therefore, it is highly unlikely that the drug alters long-term memory. Unfortunately, our university is allowing only limited number of employees to work on site at any time. Therefore, we could not conduct all the suggested experiments and opted not to include this one.

7. What are the effects of RapaBlock alone in the liver? Are the effects observed in fibrogenic gene expression markers also observed if protein levels are tested? Where the liver enzymes affected?

Answer: To address the reviewer's question, we conducted an additional experiment and measured STAT3 phosphorylation as well as Timp1 and Col4a1 expression in the liver following chronic administration of RapaBlock alone. We did not observe changes in STAT phosphorylation or in the level of liver toxicity markers (**Supplementary Figure 2**). Because of cost and antibody quality considerations, we did not test the protein levels of Timp1 and Col4a1.

8. Does Rapablock alone modify alcohol intake or blood alcohol levels?

Answer: As shown **Supplementary Figure 5**, RapaBlock does not alter alcohol intake and preference. To address the reviewer's question regarding BAC, we conducted an additional experiment in which we tested BAC after systemic administration of RapaBlock. As shown in new **Supplementary Figure 6A**, RapaBlock does not alter alcohol metabolism.

9. Although the authors acknowledge this limitation, other peripheral organs, such as the bone marrow or pancreas, should be examined to rule gain a deeper understanding of the usefulness of this binary strategy to prevent undesired side effects of chronic mTORC1 inhibition.

Answer: In addition to the protection against liver toxicity, we tested glucose tolerance suggesting that the pancreas is protected. Furthermore, we also examined the effect of RapaLink-1 with and without RapaBlock on body weight (data herein), and muscle toxicity ⁴ suggesting that other organs are protected as well. We do think that the reviewer does have a valid point, and in an ideal world we would have been happy to set another group of animals and test the long-term effects of RapaLink-1 with and without Rapablock on other organs. We plan to conduct these experiments in the future.

10. The statistical analyses are adequate but I would recommend that the authors report the effect sizes, using the appropriate statistic.

Answer: Effect size has been added to the Figure legends.

11. The authors may want to acknowledge the caveat of the absence of experiments in females.

Answer: The reviewer is absolutely correct. We apologize for this oversight. The exclusion of female mice was due to the fact that a previous study reported that rapamycin does not alter alcohol intake in female mice ¹⁰. We are currently in the process of examining whether we can replicate Cozzoli's findings. We also plan to test whether mTORC1 inhibitors alter other alcohol-dependent behaviors in female animals. The sex of the mice is now specified and we added this caveat to the discussion.

Minor comments:

12. *Figure 1 seems unnecessary to me.*

Answer: We believe that the diagram is helpful for the readers, however we keep the decision of whether or not to include the Figure to the discretion of the editor.

13. *Fig. 4C Legend: Does 'Shame' refer to VH?*

Answer: Sham was added to indicate that the water mice did not receive a systemic administration (vehicle or drug). To avoid confusion, we took the word sham out.

14. *Figure 4 A Add 'last' between 'the' and '24 h' in the text below the brain slice diagram.*

Answer: We modified the text according to the reviewer's suggestion.

Reviewer #3 (Remarks to the Author):

This report from Ehinger and colleagues provides exciting findings relating to possible pharmacological therapies for alcohol use disorder (AUD). Prior work has shown in rodent models that mTORC1 Inhibitors, such as rapamycin, reduce neurobiological impacts of alcohol and alcohol intake itself. The author and her lab have spearheaded the bulk of this prior body of work. I am familiar with this work and find it strong and compelling. However, a major drawback has prevented the extension of this work to humans. And that is that rapamycin and other mTORC1 inhibitors have clear, major negative side effects in humans when used chronically. To address this, the authors have taken an innovative approach in devising a way to block the disadvantageous peripheral effects, but not the desired central nervous system effects, of mTORC1 inhibitors by co-administering an agent that prevents peripheral action but does not cross the blood brain barrier, leaving central drug actions intact.

Through a series of animal control studies, the authors demonstrate protected liver function and glucose levels when the mTORC1 antagonist Rapalink1 is given with the blocking agent, 'rapablock', itself. This team has previously shown that Rapalink1 is an mTORC1 antagonist with similar or in some case improved attributes to rapamycin. The authors go on to use molecular biology and behavioral pharmacology in mice to characterize the impacts of these two compounds. The authors show that alcohol intake is reduced in the standard animal model of high-drinking C57 mice. The experiments are well-designed, with the proper controls, the sample sizes are sufficient for me to be convinced of the findings. I am very pleased to report that I have no major issues with the work as presented. It is a timely and significant advance that moves us closer to treatment in humans.

There are some minor issues with the writing and presentation and a few questions, listed below.

1. The following sentence will lead some to believe there are human studies in this paper. Needs better wording: We show that the dual administration of RapaLink-1 with RapaBlock, abolishes RapaLink-1- dependent mTORC1 inhibition in the liver and blocks adverse side effects detected in humans including body weight loss, glucose intolerance and liver toxicity.

Answer: We apologize for the confusion. We modified the sentence and changed the text of the manuscript to differentiate between work in rodents and human data.

2. Can the authors comment on the likely use of this drug combination chronically. The rodent experiment reported findings from an acute injection. I am not suggesting this study be added to this work as it is already an extensive investigation.

Answer: The reviewer makes a valid point that the chronic actions of rapamycin and RapaLink-1 on alcohol intake have not been tested. It is plausible that the inhibitory actions of the drugs on alcohol intake, reward, relapse seeking and habit which were tested using a single drug administration regime, could produce a long-lasting effects when given multiple times. This possibility will be tested in future studies. We added this point to the discussion.

3. Line 42, recommend change 'use' to 'intake'

4. Line 53, change semi-colon after widespread to a comma unless the wording of the following phrase is edited to become a clause.

5. Line 210: Needs rewording; Our data suggest that RapaBlock while acting in the periphery, the drug is inert in the CNS. 6. Must be clear to the reader that only males were tested; should be in abstract.

Answer: We changed the text per the reviewer's request. See also Reviewer 2 point 11.

References

- 1 Laguesse, S., Morisot, N., Phamluong, K. & Ron, D. Region specific activation of the AKT and mTORC1 pathway in response to excessive alcohol intake in rodents. *Addict Biol* **22**, 1856-1869, doi:10.1111/adb.12464 (2017).
- 2 Fan, Q. *et al.* A Kinase Inhibitor Targeted to mTORC1 Drives Regression in Glioblastoma. *Cancer Cell* **31**, 424-435, doi:10.1016/j.ccell.2017.01.014 (2017).
- 3 Morisot, N., Novotny, C. J., Shokat, K. M. & Ron, D. A new generation of mTORC1 inhibitor attenuates alcohol intake and reward in mice. *Addict Biol* **23**, 713-722, doi:10.1111/adb.12528 (2018).
- 4 Zhang, Z. *et al.* Achieving Brain-Restricted mTOR Inhibition with Binary Pharmacology. *bioRxiv* (2020).
- 5 Neasta, J., Ben Hamida, S., Yowell, Q., Carnicella, S. & Ron, D. Role for mammalian target of rapamycin complex 1 signaling in neuroadaptations underlying alcohol-related disorders. *Proc Natl Acad Sci U S A* **107**, 20093-20098, doi:10.1073/pnas.1005554107 [pii]10.1073/pnas.1005554107 (2010).
- 6 Beckley, J. T. *et al.* The First Alcohol Drink Triggers mTORC1-Dependent Synaptic Plasticity in Nucleus Accumbens Dopamine D1 Receptor Neurons. *J Neurosci* **36**, 701-713, doi:10.1523/JNEUROSCI.2254-15.2016 (2016).
- 7 Barak, S. *et al.* Disruption of alcohol-related memories by mTORC1 inhibition prevents relapse. *Nat Neurosci* **16**, 1111-1117, doi:10.1038/nn.3439 [pii]10.1038/nn.3439 (2013).
- 8 Ben Hamida, S. *et al.* Mammalian target of rapamycin complex 1 and its downstream effector collapsin response mediator protein-2 drive reinstatement of alcohol reward seeking. *Addict Biol* **24**, 908-920, doi:10.1111/adb.12653 (2019).
- 9 Morisot, N. *et al.* mTORC1 in the orbitofrontal cortex promotes habitual alcohol seeking. *Elife* **8**, doi:10.7554/eLife.51333 (2019).
- 10 Cozzoli, D. K. *et al.* Functional regulation of PI3K-associated signaling in the accumbens by binge alcohol drinking in male but not female mice. *Neuropharmacology* **105**, 164-174, doi:10.1016/j.neuropharm.2016.01.010 (2016).

Reviewer #1 (Remarks to the Author):

My comments have been addressed

Reviewer #2 (Remarks to the Author):

I have now reviewed Ms. NCOMMS-20-45438A. I see that the authors have adequately addressed all my comments and concerns and have performed additional experiments to expand and strengthen their work. I would like to commend them for such effort especially in times when the access to the laboratories may not be easy. I have no further comments and suggest acceptance in the current form